# Functional and structural analyses of novel Smith-Kingsmore Syndrome-Associated *MTOR* variants reveal potential new mechanisms and predictors of pathogenicity

Aaron D. Besterman [1,2,3,4,5,6], Thorsten Althoff [7], Peter Elfferich [8], Irma Gutierrez-Mejia [3,9], Joshua Sadik [10], Jonathan A. Bernstein [11], Yvette van Ierland [8,12], Anja A. Kattentidt-Mouravieva [13], Mark Nellist [8,12], Jeff Abramson [7], Julian A. Martinez-Agosto [1,3,9]*

1 University of California Los Angeles, Semel Institute for Neuroscience and Human Behavior, Los Angeles, California, United States of America, 2 University of California Los Angeles, Division of Child and Adolescent Psychiatry, Department of Psychiatry, Los Angeles, California, United States of America, 3 University of California Los Angeles, Division of Medical Genetics, Department of Pediatrics, Los Angeles, California, United States of America, 4 University of California San Diego Department of Psychiatry, Division of Child and Adolescent Psychiatry, San Diego, California, United States of America, 5 Rady Children's Hospital of San Diego, San Diego, California, United States of America, 6 Rady Children's Institute for Genomic Medicine, San Diego, California, United States of America, 7 Department of Physiology, David Geffen School of Medicine, University of California Los Angeles, Los Angeles, California, United States of America, 8 Department of Clinical Genetics, Erasmus Medical Center, Rotterdam, The Netherlands, 9 University of California Los Angeles, Department of Human Genetics, Los Angeles, California, United States of America, 10 University of California Los Angeles, David Geffen School of Medicine, Los Angeles, California, United States of America, 11 Department of Pediatrics, Division of Medical Genetics, Stanford University School of Medicine, Stanford, California, United States of America, 12 ENCORE Expertise Center for Neurodevelopmental Disorders, Erasmus Medical Center, Rotterdam, The Netherlands, 13 Zuidweste Foundation, Middelharnis, The Netherlands

* julianmartinez@mednet.ucla.edu

**Data Availability Statement:** All relevant data are within the manuscript and its Supporting

## Abstract

Smith-Kingsmore syndrome (SKS) is a rare neurodevelopmental disorder characterized by macrocephaly/megalencephaly, developmental delay, intellectual disability, hypotonia, and seizures. It is caused by dominant missense mutations in *MTOR*. The pathogenicity of novel variants in *MTOR* in patients with neurodevelopmental disorders can be difficult to determine and the mechanism by which variants cause disease remains poorly understood. We report 7 patients with SKS with 4 novel *MTOR* variants and describe their phenotypes. We perform *in vitro* functional analyses to confirm *MTOR* activation and interrogate disease mechanisms. We complete structural analyses to understand the 3D properties of pathogenic variants. We examine the accuracy of relative accessible surface area, a quantitative measure of amino acid side-chain accessibility, as a predictor of *MTOR* variant pathogenicity.

We describe novel clinical features of patients with SKS. We confirm MTOR Complex 1 activation and identify MTOR Complex 2 activation as a new potential mechanism of disease in SKS. We find that pathogenic *MTOR* variants disproportionately cluster in hotspots in the core of the protein, where they disrupt alpha helix packing due to the insertion of bulky

Information files. However, individual patient data beyond what is presented cannot be shared publicly due to issues of patient privacy and confidentiality. For any inquiries about data access, please contact the UCLA Office of Human Research Protection Program at 10889 Wilshire Blvd, Suite 830 Los Angeles, CA 90095. mirb@research.ucla.edu.

**Funding:** ADB was supported by the National Institute of General Medical Sciences 2T32GM008243-3 (http://nigms.nih.gov) and the American Academy of Child and Adolescent Psychiatry Pilot Research Award for Junior Faculty and Child and Adolescent Psychiatry Fellows (https://www.aacap.org/). JS was supported by the UCLA Medical Student Summer Research Fellowship Program in Psychiatry and Biobehavioral Sciences (https://medschool.ucla.edu/current-summer-research-opportunities). TA and JA were supported by the National Institute of General Medical Sciences 1R35GM135175-01 (http://nigms.nih.gov). The funders had no role in study design, data collection and analysis, decision to publish, or preparation of the manuscript.

**Competing interests:** The authors have declared that no competing interests exist.

amino acid side chains. We find that relative accessible surface area is significantly lower for SKS-associated variants compared to benign variants. We expand the phenotype of SKS and demonstrate that additional pathways of activation may contribute to disease. Incorporating 3D properties of *MTOR* variants may help in pathogenicity classification. We hope these findings may contribute to improving the precision of care and therapeutic development for individuals with SKS.

## Author summary

Smith-Kingsmore Syndrome is a rare disease caused by damage in a gene named *MTOR* that is associated with excessive growth of the head and brain, delays in development and deficits in intellectual functioning. We report 7 patients who have changes in MTOR that have never been reported before. We describe new medical findings in these patients that may be common in Smith-Kingsmore Syndrome more broadly. We then identify how these new gene changes impact the function of the MTOR protein and thus cell function downstream. Lastly, we show that changes in the gene that lie deep inside the 3D structure of the MTOR protein are more likely to cause disease than those changes that lie on the surface of the protein. We may be able to use the 3D properties of *MTOR* gene changes to predict if future changes we see are likely to cause disease or not.

## Introduction

The PI3K/AKT/MTOR signaling pathway plays a central role in cellular growth and neurodevelopment. Germline and somatic variants in genes within this signaling pathway, including *PTEN*, *PIK3CA*, *PIK3R2*, and *AKT3*, have been implicated in overgrowth and neurodevelopmental disorders (NDD), characterized by the excess growth of the body and/or brain, depending on the specific distribution and degree of activation of the variant [1,2]. More recently, activating missense variants in *MTOR* have been implicated in a range of overgrowth disorders that can strongly impact neurodevelopment [3]. MTOR plays a central role in controlling cellular growth by sensing environmental elements, such as oxygen, amino acid abundance, and growth factors, and by regulating downstream anabolic and catabolic processes through 2 multi-component protein complexes: MTOR Complex 1 (MTORC1) and MTOR Complex 2 (MTORC2) [4].

The phenotypic impact of *MTOR* variants depends largely on the degree of activation and somatic distribution of the variant [1]. Somatic *MTOR* variants have been associated with a clinical presentation of asymmetric megalencephaly (including hemimegalencephaly), polymicrogyria, and cutaneous pigmentary mosaicism, while variants confined to subregions of the brain have been associated with focal cortical dysplasia (FCD) types 2a and 2b, which frequently result in epilepsy and other neurodevelopmental deficits [1,3]. Germline and somatic mosaic *MTOR* pathogenic variants [5] cause Smith-Kingsmore Syndrome (SKS), which is characterized by macrocephaly/megalencephaly, developmental delay, intellectual disability, hypotonia, and seizures [3].

To date, more than a dozen different *MTOR* variants have been implicated in SKS, primarily localized within the focal adhesion targeting (FAT) and kinase domains of MTOR, which is where many cancer-associated *MTOR* variants are localized as well [6]. However, a large number of *MTOR* variants that map to these domains are also found in the general population.

Even with the use of consensus-based pathogenicity criteria to classify *MTOR* variants[7], it remains challenging to distinguish rare, benign population *MTOR* variants from pathogenic NDD- and cancer-associated variants. There has been a growing appreciation that the effects of missense variants on tertiary (e.g., 3D) protein structure may be predictive of pathogenicity [8–12].

The first structure of complete human MTORC1 in complex with regulatory-associated protein of MTOR (RAPTOR) and MTOR Associated Protein, LST8 Homolog (mLST8) was solved in 2016 [13] and revealed a symmetrical homodimer of rhomboid shape. MTOR largely consists of tightly packed bundles of alpha helices with a clear separation into domains. Additional structures in complex with regulating interacting proteins provided a deeper understanding of activation and inhibition of MTOR [14]. In the inactive state, the FAT-domain forms a tight clamp around the kinase domain, forcing the 2 lobes together and restricting access to the catalytic cleft. The transition from the inactive to the active state involves several twisting, rotating, and sliding motions between and within domains, particularly in the FAT domain. This causes the FAT domain to loosen its grip around the kinase so that the two lobes can rearrange and open access to the catalytic site. It has been suggested that cancer-associated mutations disturb these motions and prime MTOR into a more active state.

However, little is known about the impact of NDD-associated MTOR missense variants on protein function or structure, how they may compare to cancer-associated variants, and whether the 3D properties of variants may enhance the accuracy of predicting pathogenicity. Here, we report 7 SKS patients with 4 novel *MTOR* missense variants and describe several new clinical findings. We measure downstream activation of MTORC1 and MTORC2 in comparison to known activating and benign variants. We then perform structural analyses to better elucidate mechanisms of activation. We identify relative accessible surface area (e.g., relative solvent accessibility; RSA), a quantitative measure of amino acid side-chain accessibility [15], as a plausible predictor of pathogenicity for NDD-associated *MTOR* variants.

## Results

### Patient demographics and clinical phenotypes

We report 7 patients (5 males and 2 females) from 2 medical centers. All individuals had previously undescribed variants in *MTOR* and clinical phenotypes consistent with SKS, including at least macrocephaly and developmental delay (Table 1). The p.V2460M variant was identified in 3 patients, p.K1395R in 2 patients, and p.C1390Y and p.Q2524K present in 1 patient each (Table 1). Five of the 7 patients were 5 years of age or younger and all but 1 were of European ancestry. We report one of the oldest known SKS patients, who passed away from an accident at the age of 47 (Table 1) but who had presented with progressive cognitive decline associated with white matter changes and with 2 intraosseous meningiomas (Table 1). Four of the patients had abnormal findings on neuroimaging, with pathology of the corpus collosum in two being the only commonality. In addition to the previously reported common features of SKS [3], multiple patients had constipation, recurrent infections, asthma, an abnormal corpus collosum, and anemia (Table 1).

### Functional analysis

To estimate the effects of the *MTOR* variants on MTORC1 activity, we expressed the variants together with an S6K reporter construct in HEK 293T cells (Fig 1A–1D). All variants were expressed at approximately equal levels (Fig 1A and 1B). To compare the MTORC1 activity between variants, we determined the T389/S6K ratio in the presence of transfected *MTOR* variants relative to the ratio of transfected (T389/S6K = 1.0) wild-type MTOR. The T389/S6K

**Table 1. Summary of Clinical Features of Smith-Kingsmore Syndrome in Patients with Novel Variants.**

| Patient | Patient 1 | Patient 2 | Patient 3 | Patient 4 | Patient 5 | Patient 6 | Patient 7 |
|---|---|---|---|---|---|---|---|
| Variant | p.K1395R | p.K1395R | p.C1390Y | p.V2406M | p.V2406M | p.V2406M | p.Q2524K |
| Domain | FAT | FAT | FAT | Kinase C-Lobe | Kinase C-Lobe | Kinase C-Lobe | FATC |
| Inheritance | De Novo | De Novo | De Novo | De Novo | De Novo | De Novo | Unknown |
| Age (years) | 5 | 5 | 4 | 3 | 4 | 30 | 52 |
| Sex | Male | Male | Male | Male | Female | Male | Female |
| Ancestry | Latin American | European | European | European | European | European | European |
| Country | USA | USA | Netherlands | USA | UK | Netherlands | Netherlands |
| Deceased | - | - | - | - | - | - | Accident |
| **Core SKS Features** | | | | | | | |
| Macrocephaly/ megalencephaly | + | + | + | + | + | + | + |
| Developmental Delay | + | + | + | + | + | + | + |
| Intellectual disability | + | + | + | Undetermined | Undetermined | + | + |
| **Common SKS Features** | | | | | | | |
| Autism Spectrum Disorder | - | + | - | - | - | - | - |
| Hypotonia | + | + | + | + | + | + | - |
| Seizures | + | - | - | - | - | + | + |
| Dysmorphic Facial Features | + | + | + | + | + | + | + |
| Ventriculomegaly | + | + | - | + | - | + | + |
| **Other Clinical Features by System** | | | | | | | |
| Head and Neck | Laryngeal cleft, Type 1, Dysphagia and aspiration, OSA | | | OSA | Dysphagia, aspiration | Optic atrophy | Cataract age 47 |
| Cardiopulmonary | | Asthma | Asthma | | | | |
| Gastrointestinal | | Constipation | | | Constipation | Constipation | Constipation |
| Genito-Urinary | | | | Undescended testes | | | |
| Hematology/ Oncology | Neutropenia and anemia | | | | | Recurrent, mild anemia | |
| Neurological | | Thinning of the corpus collosum | Left cerebral hemisphere slightly larger than right hemisphere. Cavum septum pellucidum | | | Dysgenesis of the corpus collosum, thin chiasma opticum, delayed myelinization, GTC epilepsy started first month of life with new seizure type with bradycardia, low BP later onset | Seizure onset at age 47. Cognitive decline, at age 48 with periventricular white matter abnormalities, hypodense lesion parieto-occipital sulci, diffuse brain atrophy, 2 intraosseous meningiomas. |
| Psychiatric | | Poor sleep maintenance, irritability, self-injurious behaviors | | | | | Poor sleep maintenance, irritability, self-injurious behaviors |

(*Continued*)

**Table 1.** (Continued)

| Patient | Patient 1 | Patient 2 | Patient 3 | Patient 4 | Patient 5 | Patient 6 | Patient 7 |
|---|---|---|---|---|---|---|---|
| Endocrine | | | | | | Hypothyroidism, Excessive thirst | |
| Infection Disease | | Recurrent infections | Recurrent infections | | | Recurrent Infections | Recurrent infections |
| Musculoskeletal | | | | | | Bifid distal phalanx of left thumb, severe scoliosis | |
| Skin and Hair | | Dry skin, capillary malformation | | Hemangioma | | | |

BP = blood pressure, FAT = focal adhesion targeting; GTC = generalized tonic-clonic, OSA = obstructive sleep apnea; SKS = Smith-Kingsmore syndrome

ratio for all the MTOR variants observed in our patients (p.C1390Y, p.K1395R, p.V2406M, and p.Q2524K) increased 1.5-fold to less than 5-fold relative to transfected wild-type MTOR (Fig 1A and 1C). These differences were statistically significant ($p < 0.05$, Student's paired $t$-test with Bonferroni correction). Similar results were obtained for the previously reported activating, NDD-associated variants (p.C1483F, p.E1799K) and cancer-associated variants (p. V2406A, p.E2419K, p.Q2524L) [6,16,17]. In contrast, the T389/S6K ratios in the presence of the MTOR p.P2522A and p.V2525I benign variants found in gnomAD were not significantly different to transfected wild-type MTOR ($p < 0.05$, Student's paired $t$-test with Bonferroni correction). Expression of the TSC complex, a negative regulator of MTORC1, reduced the T389/S6K ratio, which was consistent with previous studies (15). Unexpectedly, the T389/S6K ratio in cells transfected with empty vector had an approximately 3-fold increase relative to cells transfected with wild-type *MTOR* ($p < 0.05$, Student's paired $t$-test with Bonferroni correction) (Fig 1A and 1C).

To estimate the effects of the MTOR variants on MTORC2 activity, we expressed the variants together with an AKT3 reporter construct in 3H9-1B1 (TSC1:TSC2 KO) cells (Fig 1E–1H). All transfected variants were expressed at approximately equal levels (Fig 1E and 1F). To compare the MTORC2 activity between variants, we determined the S473/AKT ratio in the presence of the transfected MTOR variants relative to transfected wild-type MTOR (Fig 1E and 1G). The p.C1483F variant, a previously reported highly activating variant [6,16] (Table 2), was associated with an approximate doubling of S473/AKT. The degree of activation of MTORC2 by the novel variants was more modest. S473/AKT increased approximately 1.5-fold compared to transfected wild-type for p.Q2524K and 1.2-fold for p.C1390Y and p. V2406M. Due to a larger number of replicates for p.C1390Y, it was found to be significantly greater than the transfected wild-type MTOR ($p < 0.05$, Student's paired $t$-test with Bonferroni correction), while p.V2406M was not ($p = 0.06$). The S473/AKT ratios for the p.K1395R and p. V2406M transfected variants were not significantly different from wild-type MTOR.

## Structural analysis

To further assess the variants identified in our SKS patients and better understand their functional implications, we analysed the structure of MTORC1 (PDB 6BCX). All 4 variants are in the core the protein, either in the FAT or kinase domains. These variants lie at the domain interfaces where amino acid side chains are close enough to interact with each other (Fig 2). This is a common feature for most NDD- and cancer-associated variants (Fig 2). In contrast, benign population variants reside predominantly in the Huntington, Elongation Factor 3, PR65/A, TOR (HEAT) domains and on the exterior, solvent-exposed surface of MTOR.

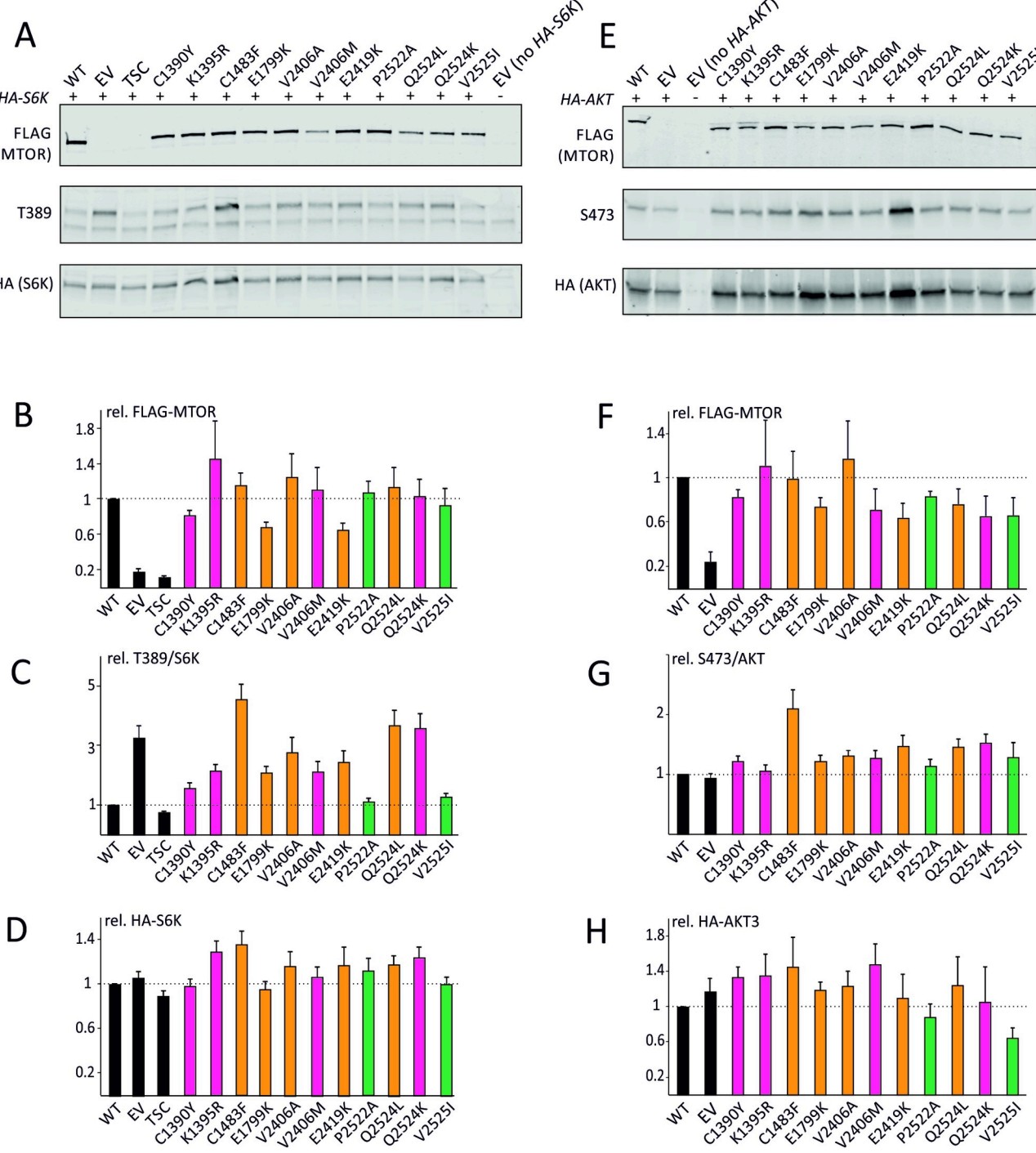

**Fig 1. Novel Smith-Kingsmore Syndrome Variants Activate MTORC1 and MTORC2 in Stimulated and Serum-starved Human Embryonic Kidney Cells.** (A) An example of an immunoblot is shown. (B) Expression levels of the FLAG-MTOR variants were determined using an antibody specific for the FLAG epitope. (C) To estimate TORC1 activity, the ratio of T389-phosphorylated S6K (T389) to total HA-S6K was determined, relative to the ratio for wild-type MTOR (T389/S6K; = 1.0). (D) The total HA-S6K signal was used to estimate the transfection efficiency. (E) An example of an immunoblot is shown. (F) Expression levels of the FLAG-MTOR variants were determined using an antibody specific for the FLAG epitope. (G) To estimate MTORC2 activity, the ratio of S473-phosphorylated AKT3 (S473) to total HA-AKT3 was determined, relative to the ratio for wild-type MTOR (S473/AKT = 1.0). (H) The total HA-AKT3 signal was used to estimate the transfection efficiency.

**Table 2. Summary of NDD-associated *MTOR* Variants.**

| HGVS Coding Variant | HGVS Protein Variant | SKS | FCD 2a/2b | AM/PMG/CPM | Domain | Interface Location | pS6K (T389) Activation (MTORC1) | pAKT (S473) Activation (MTORC2) |
|---|---|---|---|---|---|---|---|---|
| c.1871G>A | p.R624H | | [26] | | N-Heat | | | |
| c.4169G>A | p.C1390Y | Current Study | | | FAT | | Current Study | Current Study |
| c.4184A>G | p.K1395R | Current Study | | | FAT | FAT Hinge | Current Study | Current Study |
| c.4348T>G | p.Y1450D | | [26] | | FAT | FAT Hinge | | |
| c.4487T>G | p.W1456G | | [27] | | FAT | FAT Hinge | | |
| c.4376C>A | p.A1459D | | [28,29] | | FAT | FAT Hinge | | |
| c.4375G>T | p.A1459S | | [30] | | FAT | FAT Hinge | | |
| c.4379T>C | p.L1460P | | [25,28–31] | | FAT | FAT Hinge | [6,16,17,25] | [6] |
| c.4447T>C | p.C1483R | | [26,31] | | FAT | FAT Hinge | [25,26] | |
| c.4448G>T | p.C1483F | [32] | | | FAT | FAT Hinge | [6,16] | [6] |
| c.4448C>T | p.C1483Y | | | [31,33–35] | FATC | FAT Hinge | [25] | |
| c.4468T>C | p.W1490R | [30] | | | FAT | FAT Hinge | | |
| c.4785C>T | p.M1595I | [30] | | | FAT | | | |
| c.5005G>T | p.A1669S | | | [35] | FAT | | | |
| c.5126G>A | p.R1709H | | [26] | | FAT | FAT Hinge | | |
| c.5395G>A | p.E1799K | [3,25,33,34,36–39] | | | FAT | N-lobe C-lobe Interface | [6,25] | [6] |
| c.5494G>A | p.A1832T | [30] | | | FAT | | | |
| c.5663A>C | p.F1888C | [30] | | | FAT | N-lobe C-lobe Interface | | |
| c.5930C>G | p.T1977R | | [31] | | FAT | FAT-N-lobe Interface | [6] | [6] |
| c.5930C>T | p.T1977I | | | [25,40] | FAT | FAT-N-lobe Interface | [25] | |
| c.5930C>A | p.T1977K | | [26,29] | [31] | FAT | FAT-N-lobe Interface | [16] | |
| c.6577C>T | p.R2193C | | [26] | | Kinase N-Lobe | | | |
| c.6644C>A | p.S2215F | | [25,26,28–30] | [31] | Kinase N-Lobe | N-lobe C-lobe Interface | [16,25] | |
| c.6644C>T | p.S2215Y | | [25,28–31] | [31] | Kinase N-Lobe | N-lobe C-lobe Interface | [6,17,25] | [6,17] |
| c.6605T>G | p.F2202C | [3] | | | Kinase N-Lobe | N-lobe C-lobe Interface | | |
| c.6981G>A | p.M2327I | [30] | | | Kinase C-Lobe | ATP Binding Site | [16] | |
| c.7216G>A | p.V2406M | Current Study | | | Kinase C-Lobe | N-lobe C-lobe Interface | Current Study | Current Study |
| c.7235A>T | p.D2412V | [41] | | | Kinase C-Lobe | N-lobe C-lobe Interface | | |
| c.7280T>A | p.L2427Q | | [26] | | Kinase C-Lobe | ATP Binding Site | [26] | |
| c.7280T>C | p.L2427P | | [26] | | Kinase C-Lobe | ATP Binding Site | [26] | |
| c.7498A>T | p.I2500F | [29] | | | Kinase C-Lobe | | | |
| c.7501A>G | p.I2501V | [30]* | | | Kinase C-Lobe | | | |
| c.7570C>A | p.Q2524K | Current Study | | | FATC | | Current Study | Current Study |

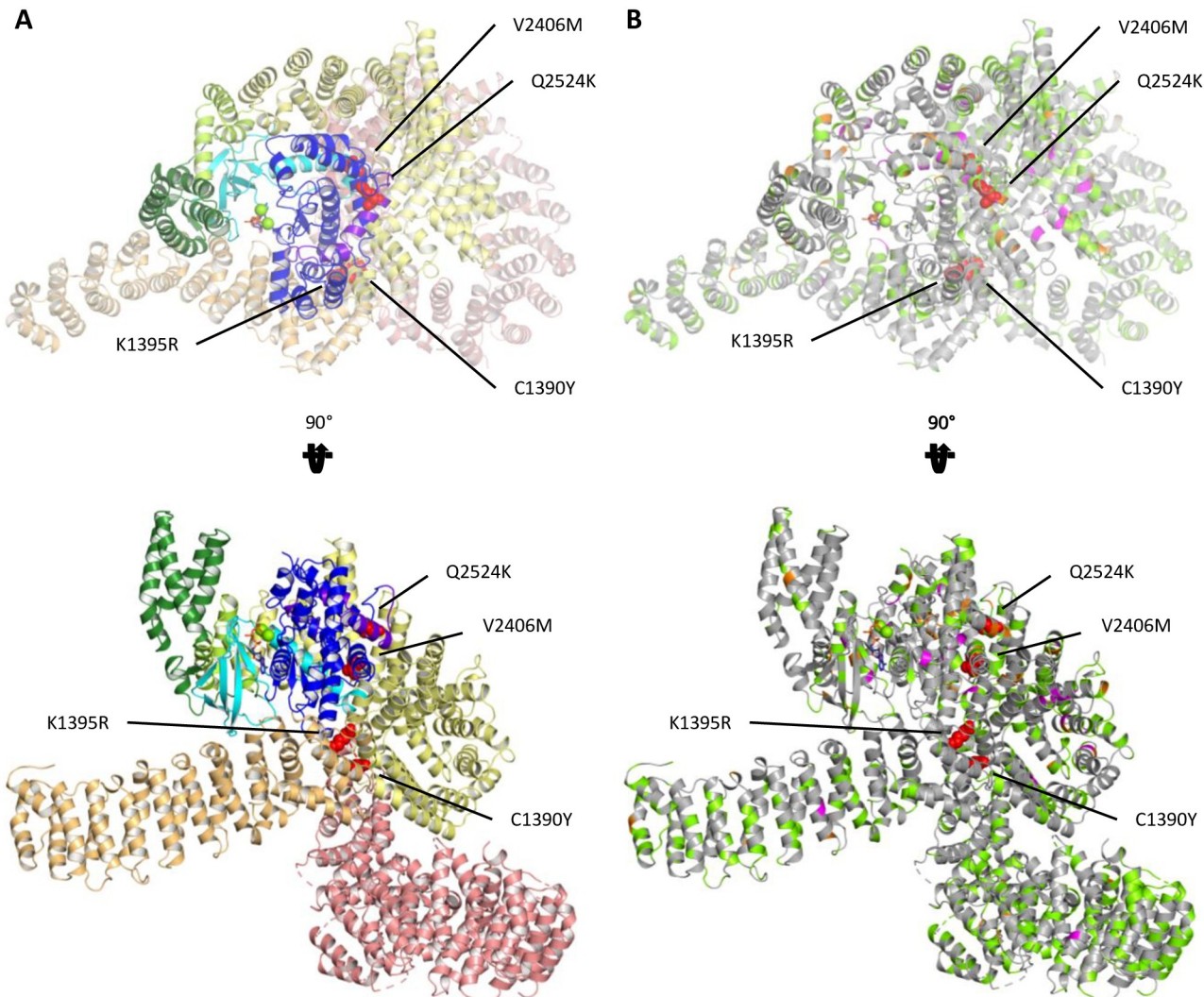

**Fig 2. Cryogenic Electron Microscopy Structure of MTOR with Mapped Novel Variants.** Overview of MTOR structure (derived from Protein Data Bank 6BCX) as seen from "top" and from one side after a 90˚ rotation around the X axis. (A) Domain coloring as follows: N-HEAT, salmon; M-HEAT, light orange; FAT, yellow; FAT-kinase linker, light green; FKBP-rapamycin-binding (FRB), dark green; kinase N-lobe, cyan; kinase C-lobe, blue; FRAP, ATM, TRRAP C-terminal (FATC), purple. (B) Mutated residues from the gnomAD database are highlighted in bright green. Mutated residues from the COSMIC database are highlighted in orange. Mutations associated with the phenotypes for neurodevelopmental disorder (NDD) are highlighted in magenta. New overgrowth mutations identified in this study are shown as red spheres.

In all cases of pathogenic variants, a short or more compact amino acid side chain (cysteine, valine, glutamine, lysine) was replaced with a longer and bulkier side chain (tyrosine, methionine, lysine, or arginine, respectively). Even though at 3.0 Å the resolution of the structure is a bit too low to confidently model the positions of all amino acid side chains and we acknowledge that certain side chain rotamers might be accommodated slightly better than others, it is clear that these modifications cannot be easily accommodated in the tightly packed α-helices without distorting the neighboring helices (Fig 3). In general, pathogenic *MTOR* variants cluster in 4 locations: around residues 1483, 1799, 1977 and 2215 (S2 Fig). These clusters are in close proximity to interface regions critical for allosteric activation of MTOR like the FAT-hinge or between the 2 lobes of the kinase domain. In these mutational hotspots, the disease-associated mutations face toward neighboring helices, whereas the benign population variants

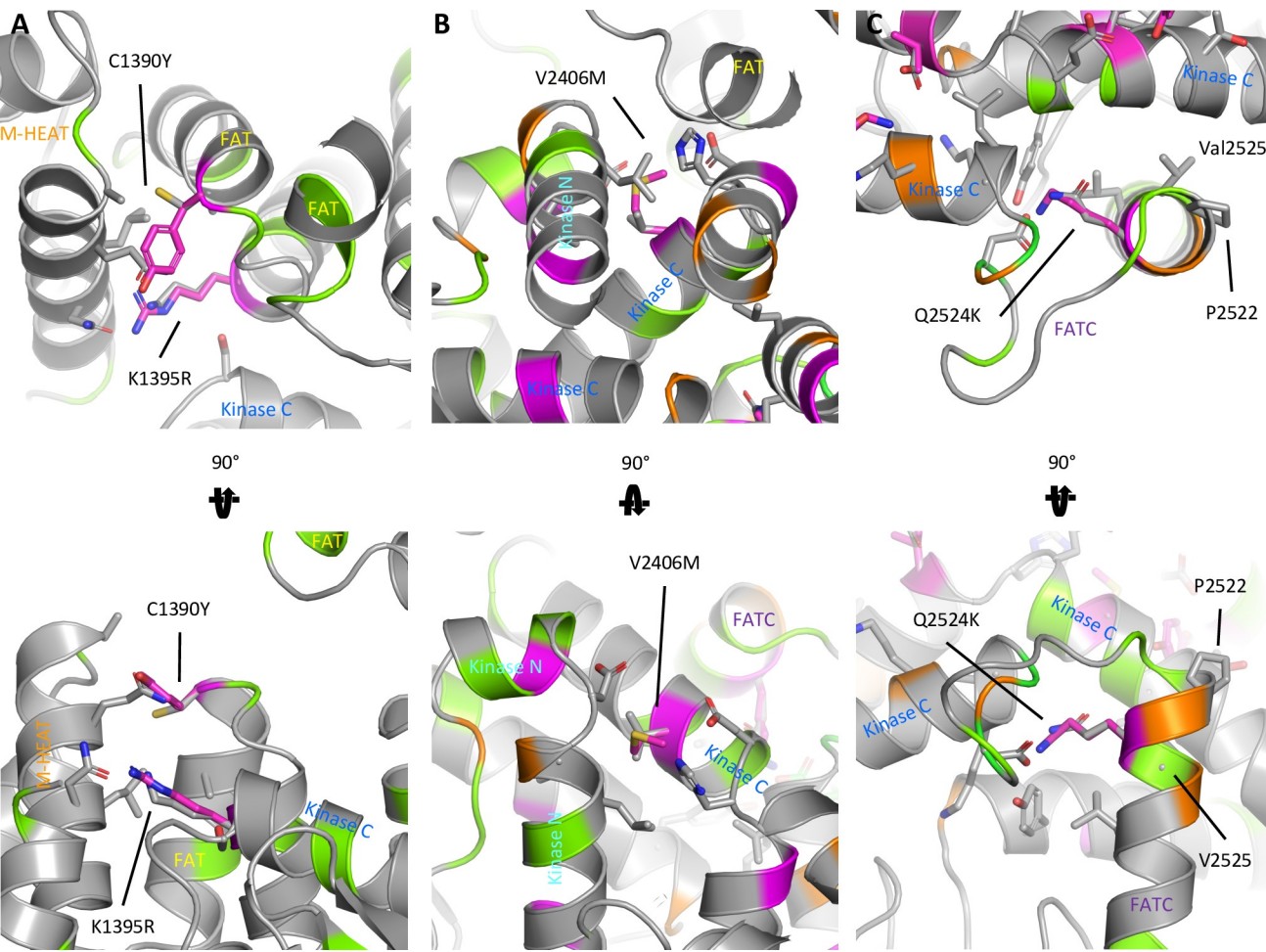

**Fig 3. Pathogenic MTOR Variants Cluster in Key Tertiary Protein Structures and Lie on Internally Facing Helical Surfaces.** Detailed view of mutations investigated in this study. (A) p.C1390Y, p.K1395R. (B) p.V2406M, (C) p.Q2524K. Wild-type side chains are shown as gray sticks. Mutated side chains are shown as magenta sticks. Neighboring residues in close proximity are shown as gray sticks. Variants associated with the phenotypes for neurodevelopmental disorder are shown in magenta; cancer variants from the COSMIC database, in orange; and population variants from the gnomAD database, in green.

mapping to residues immediately adjacent in the secondary structure face toward the surface of the protein, where changes to the side chain are more easily accommodated.

## RSA analysis

We sought to confirm our structural analysis findings by comparing a quantitative measure of residue accessibility (e.g., how deep or superficial an amino acid residue is in the 3D protein structure) between disease- and population-associated variants. RSA is a quantitative measure of solvent accessibility of individual side chains within a structure that can be determined for both the active and inactive states of proteins [15], where the consensus of the 2 states, or the difference between them, may also be informative.

Initially, we compared active, inactive, consensus, and differential RSA values between population and disease variants to determine the RSA measure that most strongly differentiated the groups. A log (x+1) transformation was performed to address the RSA values equal to zero and to minimize skewing. Inactive, active, and consensus RSA had similar values across

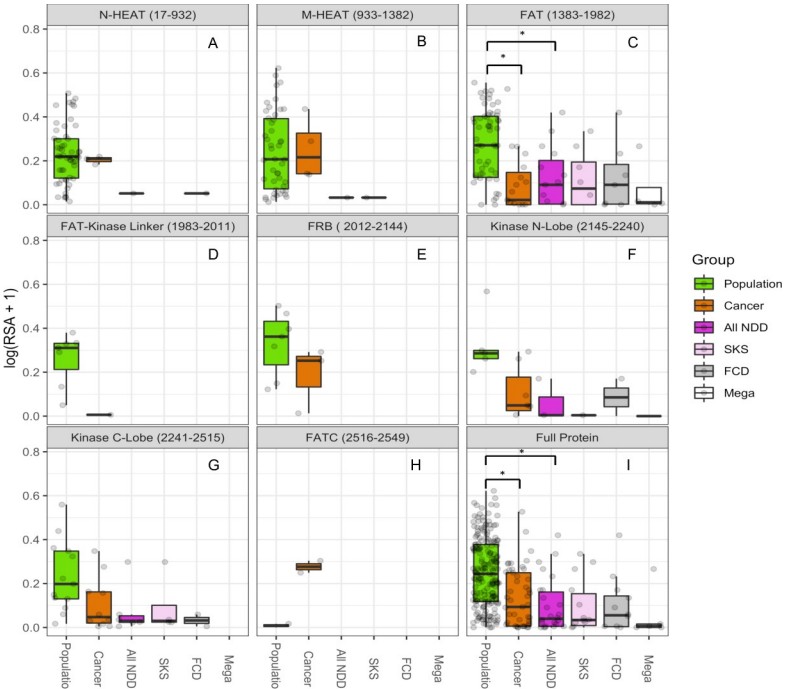

**Fig 4. Disease-associated MTOR Variants Have Lower Relative Accessible Surface Area Than Proximal Population Variants Based on Domain Location.** (A-B) and (D-H) relative accessible surface area (RSA) of MTOR variants is not significantly different between groups for these domains. (C) There is significantly lower (*, $p < 0.01$, Pairwise Wilcoxon Rank Sum Test with Bonferroni correction) RSA in disease-associated *MTOR* variants within the focal adhesion targeting (FAT) domain and (I), across the full protein. The gray panel headings contain protein domain name with residue range in brackets. SKS, FCD, and Mega are subgroups within the "All Neurodevelopmental Disorder" (NDD) group. SKS = Smith-Kingsmore syndrome; FCD = focal cortical dysplasia; Mega = megalencephaly.

groups, whereas differential RSA was uniform across groups (S3 Fig). Because there was no significant overall difference in RSA for the investigated variant residues between the active and inactive structures, all RSA analyses were performed using the consensus RSA, consistent with recent work investigating the utility of RSA to predict variant pathogenicity [10]. From here on, we refer to consensus RSA, simply as "RSA."

Using the Kruskal-Wallis test, we observed that median RSA was significantly different between population, cancer, and NDD variants ($X^2 = 39.1$, df = 2, $p = 3.2 \times 10^{-9}$) across the entire protein (Fig 4I). We also observed similar differences within the FAT domain ($X^2 = 18.6$, df = 2, $p = 9.3 \times 10^{-5}$), but much smaller differences within the kinase N lobe ($X^2 = 7.0$, df = 2, $p = 0.03$), and kinase C lobe ($X^2 = 7.0$, df = 2, $p = 0.03$) (Fig 4). Based on post-hoc analyses using the Pairwise Wilcoxon rank sum test with Bonferroni correction, we found that the median consensus RSA was significantly different between population variants and NDD variants ($p = 6.8 \times 10^{-6}$) and population and cancer variants ($p = 5.8 \times 10^{-6}$), with no difference between NDD and cancer variants ($p = 1$) across the protein (Fig 4I) Significant post-hoc differences were also observed within the FAT domain between population and NDD variants ($p = 0.01$) and population and cancer variants ($p = 7.4 \times 10^{-4}$).

However, in the M-HEAT and N-HEAT domains, which are far from the core catalytic site of the protein and lie more on the exterior surface (Fig 2), there was little difference in RSA between population and disease variants (Fig 4), consistent with the idea that there may be a different mechanisms of pathogenicity besides internal disruption of helix packing, such as disruption of binding to partner proteins [6,16].

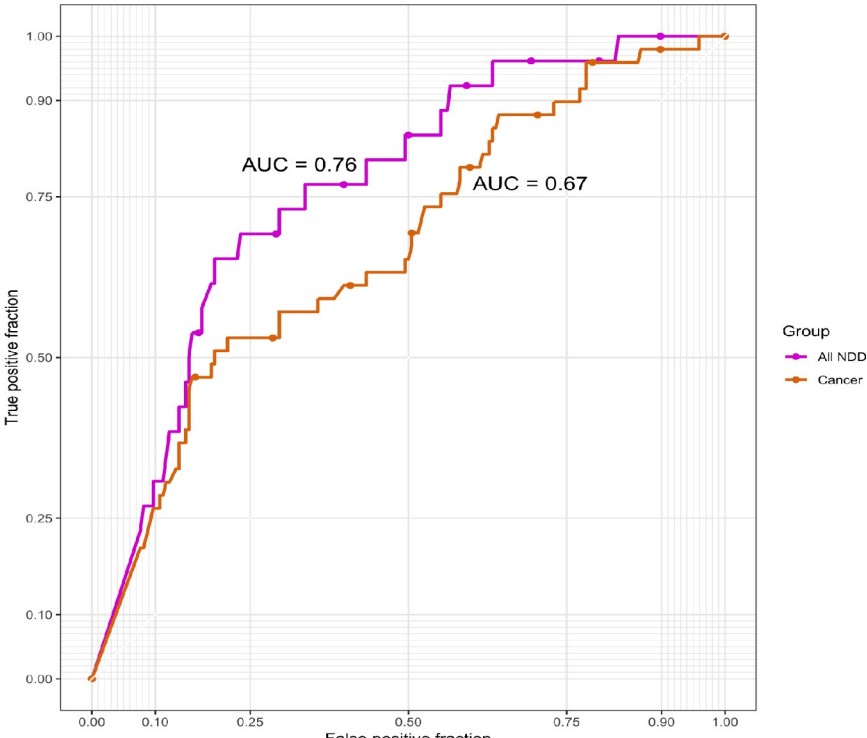

**Fig 5. ROC Curve for Relative Accessible Surface Area as a Predictor of *MTOR* Variant Pathogenicity in Neurodevelopmental Disorders and Cancer.** Receiver operating characteristic (ROC) curves were created for RSA in all neurodevelopmental disorder and cancer *MTOR* variants in comparison to population variants. AUC = area under the ROC curve.

Finally, we created a ROC curve (Fig 5) and calculated the AUC to assess the accuracy of RSA for predicting the outcome of a *MTOR* variant as being disease- or population-associated. We found that AUC = 0.76 for NDDs and 0.67 for cancer.

## Discussion

We report 4 novel autosomal dominant missense *MTOR* variants in 7 individuals with SKS. The clinical phenotype of our patients is similar to what has been previously published [3], with most having macrocephaly/megalencephaly, developmental delay, intellectual disability, facial dysmorphism, and hypotonia. Additionally, multiple patients in this study reported constipation, recurrent infections, asthma, an abnormal corpus collosum, and anemia. All of these features, except corpus collosum abnormalities, are common in both the general population and the population with autism spectrum disorder and might be incidental findings, but it is possible that overactive MTOR could increase the risk for these comorbidities as *MTOR* is known to play a key role in normal immune function and dysfunction in autism spectrum disorder [18]. Furthermore, *MTOR* activation is required for asthma onset [19], and when MTOR is inhibited pharmacologically, it often induces diarrhea [20].

We also report what may be the oldest known patient with SKS who presented with significant functional and cognitive decline from the fourth decade of life with white matter changes, diffuse brain atrophy, new-onset seizures, and 2 intraosseous meningiomas, a rare subtype representing only 2% of meningiomas [21]. The cognitive and functional decline may reflect an early-onset neurodegenerative process that may be secondary to abnormal *MTOR*

activation, as seen in Alzheimer-like dementia in Down syndrome [22]. However, additional longitudinal natural history studies of SKS patients into midlife will be necessary to confirm that this cognitive decline is associated with SKS and not incidental to this one patient. While there were neuroimaging abnormalities in other patients, such as hemispheric asymmetry in patient 3 and thinning of the chiasma opticum in patient 6 (Table 1), there was no consistent pattern of abnormalities across all patients suggesting a specific progression of neurodegenerative changes. In fact, these observations suggest significant heterogeneity in gross nervous system pathology in SKS. The intraosseous meningiomas may also represent a sporadic event or could be reflective of a cancer predisposition, which is well-known in other overgrowth disorders [2]. This is consistent with the central role that *MTOR* mutations can play in oncogenesis [16], but has yet to be demonstrated in SKS. This finding again points to the necessity for longitudinal natural history studies for SKS.

MTORC1 activation plays a central role in regulating cellular growth through protein synthesis, lipid and nucleotide synthesis, and autophagy [4]. To investigate the impact of the variants on *MTOR* signaling, we first conducted *in vitro* assays to confirm the elevated MTORC1 activity as shown in previous studies (Table 2). Unexpectedly, we found that exogenous expression through transfection of wild-type MTOR resulted in *decreased* MTORC1, but not MTORC2, activity compared to cells transfected with empty vector (Fig 1A, 1C, 1E and 1G). This is likely due to significant feedback inhibition of MTORC1, but not MTORC2, signalling, consistent with previous findings [23,24]. Nonetheless, compared to transfection with wild-type MTOR, we observed consistent MTORC1 activation when NDD- and cancer-associated *MTOR* variants were expressed, suggesting that the increase in protein activity due to the pathogenic variants exceeds the feedback inhibition secondary to increased quantity of transfected MTOR, resulting in elevated MTORC1 activity overall. This is consistent with previous studies that have observed MTORC1 activation from NDD-[25] and cancer-associated *MTOR* variants [6,16,17].

NDD-associated *MTOR* variants also resulted in increased AKT S473 phosphorylation, an indicator of MTORC2 activation. MTORC2 is responsible for cell survival and proliferation [4] and MTORC2 activation has previously been shown for cancer-associated *MTOR* variants [6,17] (Table 2), but not, to our knowledge, for NDD-associated variants. These findings have important implications for potential pharmacotherapies for SKS. Treating patients with MTORC1-only inhibitors, such as sirolimus or everolimus, may be insufficient to normalize cellular growth, and could induce positive feedback, further promoting proliferation through MTORC2-AKT signaling [24,42,43]. Recent anecdotal evidence of SKS patients trialed on off-label sirolimus (personal communication with families and clinical colleagues) suggests mixed results, with only some patients reporting clinical benefit and others reporting poor tolerability. Formalized clinical trials will be essential to establish clinical efficacy and to identify potential adverse outcomes of any precision treatments. Alternative treatment strategies may include AKT inhibitors that have shown promise in the related overgrowth disorder of Proteus Syndrome [44], dual MTORC1/2 inhibitors that are in development in oncology [45], or antisense oligonucleotide therapies directed towards *MTOR* transcripts [46].

After establishing that the novel *MTOR* variants identified in our patients were activating, we sought to better understand their mechanism of pathogenicity. Current American College of Medical Genetics guidelines suggest an approach on how best to determine variant pathogenicity by using metrics such as sequence conservation, predicted impact on protein structure, and inheritance pattern, among others [7]. Even with best-practice approaches, there are still many instances where variant pathogenicity remains difficult to determine. One approach that is being increasingly pursued is 3D structural analysis. There is mounting evidence that the 3D location of the variant and its impact on protein folding, function, and interactions, can help

predict variant pathogenicity [8,10,47,48]. It has recently been demonstrated that this approach can improve pathogenicity classification for missense variants throughout the genome [10], and more specifically for variants in *PTEN* [10–12], a protein phosphatase that is an upstream modulator of *MTOR* and also associated with NDDs and overgrowth.

We demonstrate that pathogenic *MTOR* variants, in both NDDs and in cancer, likely activate proteins by indirectly increasing exposure to the catalytic site. This may occur through disruption of alpha-helix packing in the inhibitory FAT domain (Figs 2 and S2), causing it to relax its grip around the kinase domain. In a similar fashion, they could also disturb the packing or orientation of the N- and C-lobe of the kinase domain. The pathogenic variants tend to cluster in hotspots buried in the core of the protein or, less often, lie in exterior regions of the protein that serve as interfaces for protein interaction (Figs 2 and S2), such as with the binding partners RAPTOR and DEPTOR [6,16]. Unfortunately, these parts of the protein are poorly resolved and the exact location of variant residues in those presumed interfaces remains somewhat elusive. Mutations in core regions of the protein often lie at interdomain interfaces or hinge regions where they likely interfere with the motions associated with the transition between the inactive and active conformations of the protein and priming the kinase towards a more active state (Figs 2 and S2) [14]. While mutations could also affect protein expression, our results from functional studies indicate that most variants express at levels comparable to wild-type protein (Fig 1).

We used RSA, a quantitative measure of amino acid side-chain accessibility [15], to assess differences between NDD-, cancer-, and population-associated variants protein-wide and by protein domain (Fig 4). We found that NDD-associated variants had significantly lower RSA values compared with cancer and population variants, especially in the critical FAT and kinase domains (Fig 4), suggesting that low RSA values may be predictive of pathogenicity for NDD-associated MTOR variants, as has been observed in PTEN [11,12]. We also observed that for the NDD-associated variants, megalencephaly-associated variants persistently had the lowest RSA values, which may be due to the fact that they are always present in a mosaic distribution, suggesting that more-severe protein disruption can only be tolerated in a mosaic state [1]. Based on ROC curve and AUC calculations (Fig 5), RSA has an "acceptable" [49] level of accuracy in distinguishing between pathogenic and population variants for both NDDs (AUC = 0.76) and cancer (AUC = 0.67). This level of discrimination is not clinically meaningful, which could be due to the still comparatively low number of known NDD mutations. Our analysis adds to mounting evidence that RSA may be a helpful predictor of pathogenicity for PI3K/AKT/MTOR pathway genes, or NDD-associated genes more generally. However, more work on diverse proteins is required to fully establish the predictive value of RSA and its correlation with the severity of the impact of a variant on protein function.

In summary, we report 7 SKS patients with 4 novel *MTOR* variants and describe several new clinical findings that may offer prognostic information to families, once confirmed with longitudinal natural history studies. We demonstrate that SKS-associated *MTOR* variants likely activate downstream signaling through MTORC2, in addition to MTORC1. Through structural analysis, we show that SKS-associated amino acid substitutions in MTOR disrupt alpha-helix packing in the inhibitory FAT and kinase domains and pathogenic variants tend to cluster in hotspots buried in the core of the protein. We further identify RSA as a plausible predictor of pathogenicity for NDD-associated *MTOR* variants.

## Materials and methods

### Ethics statement

The University of California Los Angeles IRB approved our study protocol (12–000989). Written informed consent was obtained from all research participants.

## Patient recruitment and assessment

Patients were recruited from 2 sites: The University of California Los Angeles Medical Center, Los Angeles, United States and the Erasmus University Medical Center (EMC), Rotterdam, The Netherlands. Participants at University of California Los Angeles were given the opportunity for a full in-person medical and neurobehavioral evaluation or an at-home questionnaire with medical records, photos, and genetic test results sent in for review. All questionnaires were reviewed, scored, and entered into a central database by I.G.M. Medical information was extracted, reviewed, and tabulated by I.G.M. and A.D.B. Patients from the Erasmus Medical Center attended the Clinical Genetics outpatient clinic. Phenotypic information was gathered from the medical records and through direct communication with the families. The patients in this study had novel *MTOR* variants, which were not previously reported to be associated with overgrowth or NDDs. The variants were classified by certified laboratories based on the American College of Medical Genetics standards and guidelines for the interpretation of sequence variants [7].

## Functional analysis

Expression constructs encoding *MTOR* variants were derived by site-directed mutagenesis (SDM) of a wild-type *MTOR* expression construct (Addgene #26603). In each case, the complete open reading frame of the construct was sequenced to ensure that no additional nucleotide changes were introduced during the SDM procedure

To assess MTORC1 activity, expression constructs encoding the *MTOR* variants were transfected into human embryonic kidney (HEK) 293T cells, together with an S6K reporter construct. Cells were maintained in a growth medium (DMEM containing 4.5 g/l D-glucose + 10% fetal calf serum) and harvested 48 hours after transfection in lysis buffer (50 mM Tris-HCl [pH 7.6], 100 mM NaCl, 50 mM NaF, 1% v/v Triton X100 and Roche cOmplete Mini Protease Inhibitor Cocktail (Roche Diagnostics GmbH, Mannheim, Germany), and the cell lysates subjected to immunoblotting. To estimate MTORC1 activity in the presence of the different variants, the cell lysates were subjected to immunoblotting and the ratio of T389-phosphorylated S6K to total S6K (T389/S6K) was determined, relative to the ratio for wild-type MTOR, similar to previous reports [6,16,17]. Five or more immunoblotting replicates were completed for each variant (S1 and S2 Data).

To assess MTORC2 activity, expression constructs encoding the *MTOR* variants were transfected into 3H9-1B1 cells, together with an AKT3 reporter construct. 3H9-1B1 cells are a HEK 293T-derived cell-line in which *Tuberous sclerosis 1 (TSC1)* and *Tuberous sclerosis 2 (TSC2)* have been inactivated by CRISPR/Cas9 gene editing [50]. The low endogenous levels of AKT-S473 phosphorylation in these cells is due to a negative feedback signalling loop, making detection of the acute effects of *MTOR* expression easier to distinguish from background signals (S1 Fig). After 24 to 48 hours, the transfected cells were harvested, and the cell lysates were subjected to immunoblotting. To estimate MTORC2 activity in the presence of the different variants, the ratio of S473-phosphorylated AKT3 to total AKT3 (S473/AKT) was determined, relative to the ratio for wild-type MTOR. Five or more immunoblotting replicates were completed for each variant.

## 3D Structural analysis

Structures with the Protein Data Bank accession codes 6BCX and 6BCU (http://www.rcsb.org) were used for this analysis as they currently represent the most comprehensive and best-resolved MTORC1 structures. For our analysis, only a single copy (chain A) was incorporated. Structures were analyzed in PyMOL (Version 2.4.0., Schrödinger Inc., New York, New York) by mutating the residues in silico and choosing the side-chain rotamer that visually caused the

least clashes with neighboring residues. Interdomain interfaces were broadly defined as areas between domains where side chains come closer than 3.0 Å. Consecutive stretches of primary sequence were selected, even if the residues were not facing toward the neighboring domain. Because of the tight packing of helices in MTOR, even mutations on the opposite side of a helix would likely distort the domain–domain interface.

## RSA analysis

All analyses were performed in R Studio (Version 1.2.1335, 2009–2019 RStudio, Inc., Boston, Massachuttes) using the tidyr package [51]. Consensus RSA values for all MTOR residues were downloaded from the MISCAST database Variant Analysis Suite on April 25, 2020 (http://miscast.broadinstitute.org/) [52]. To calculate accessible surface area (ASA) for individual MTOR structures, monomers of chain A of structures with Protein Data Bank codes 6BCX and 6BCU (activated by Ras homolog enriched in brain) were generated. The ASA was calculated using the Definition of Secondary Structure of Proteins server (https://www3.cmbi.umcn.nl/xssp/) [53,54] and converted to RSA using the theoretical maximum ASA [55]. For this analysis, a dimer of MTOR (chains A and B) was generated, but only chain A was used for further analysis. Unresolved residues that were assigned an RSA value of "-1" and residues with a missing value were excluded from the analysis.

Population-based *MTOR* variants were obtained from the genome aggregation database (gnomAD) on January 24, 2020 (https://gnomad.broadinstitute.org/) [56]. Variants present in only a single individual were removed due to the risk that they might actually be rare, pathogenic (e.g., oncogenic) variants and not benign. Variants that had been associated with overgrowth or cancer in other databases were removed. We designate these variants as "population" variants from here on.

Cancer-associated *MTOR* variants were downloaded from the COSMIC database on January 24, 2020 (https://cancer.sanger.ac.uk/cosmic) [57]. Variants that were present 3 or more times in COSMIC and were absent from gnomAD were included in our "cancer" group, to increase the likelihood that the variants were truly oncogenic. The Kruskal-Wallis test was conducted to examine the differences in RSA between population, NDD, and cancer groups. The NDD subgroups (e.g., SKS, FCD, and hemimegalencephaly) were not included in the statistical analyses, as these individuals were included in the "All NDD" group. Post-hoc, pairwise comparisons were performed using Wilcoxon rank sum tests with Bonferroni corrections for multiple comparisons. Lastly, using the plotROC R package [58], we created receiver operating characteristic (ROC) curves and calculated the area under the curve (AUC) of RSA in NDDs and cancer.

## Literature review and summary

We completed a review of the literature on July 10, 2020 to produce a comprehensive list of published *MTOR* variants associated with overgrowth disorders. We initially searched "overgrowth" AND "MTOR," each with "FCD," "Smith-Kingsmore syndrome," "megalencephaly," and "macrocephaly," respectively, in both PubMed (https://pubmed.ncbi.nlm.nih.gov/) and Google Scholar (https://scholar.google.com). We reviewed the search results for reported cases of individuals with *MTOR* variants and overgrowth phenotypes. References and citing studies were reviewed for additional published cases. A new literature search tool, "CoCites" [59] was used to identify additional citations that are frequently co-cited.

## Supporting information

**S1 Fig. TSC1 and 2 Double Knockout Results in Reduced AKT Activation.** (A and B) 3H9-1B1 (HEK 293T TSC1:TSC2 DKO) cells exhibit reduced AKT activation. TSC1:TSC2 Double

Knockout (DKO) cells were compared to the parental HEK 293T line by immunoblotting. Under conditions of nutrient and/or growth factor restriction, activated S473-phosphorylated AKT is down-regulated in the TSC1:TSC2 DKO cells, compared to the HEK 293T line. In contrast, under the same conditions, ribosomal protein S6, a marker for MTORC1 activation, is constitutively phosphorylated (pS6-S235) in the TSC1:TSC2 DKO cells. * = background band from non-specific binding.
(TIF)

**S2 Fig. NDD- and cancer-associated MTOR variants form distinct clusters in which disease-associated variants face neighboring helices, while population variants face into solvent.** Detailed view of other mutation clusters associated with NDD phenotypes. (A) p.C1483 mutation cluster. (B) p.E1799 mutation cluster. C. p.T1977I/K/R. D. p.S2215F/Y. Wild-type side chains are shown as gray sticks, mutated side chains are shown as magenta sticks. Neighboring residues in close proximity are shown as gray sticks. Mutations associated with overgrowth phenotypes are shown in magenta, mutants from the COSMIC database in orange and SNPs from the gnomAD database in green.
(TIF)

**S3 Fig. A Comparison of Active, Inactive, Consensus, and Differential RSA Measures Between Population-, Cancer- and NDD-associated MTOR Variants.** The gray panel headings indicate RSA values for the active, inactive, consensus of all structures, and differential between active and inactive, protein states. SKS, FCD and Mega are subgroups within the "All NDD" group. RSA = Relative Accessible Surface Area; NDD = Neurodevelopmental Disorder; SKS = Smith-Kingsmore Syndrome; FCD = Focal Cortical Dysplasia; Mega = Megalencephaly.
(TIF)

**S1 Table.** *MTOR* **Missense Variants Subjected to** *in vitro* **Functional Assessment.**
COSMIC = Catalogue of Somatic Mutations in Cancer; gnomAD = The Genome Aggregation Database; MTOR = mechanistic target of rapamycin; N/A = Not Applicable; SKS = Smith-Kingsmore Syndrome.
(DOCX)

**S1 Data. Data from Immunoblotting Experiments for MTORC1-S6K Activation (Fig 1).**
(XLSX)

**S2 Data. Data from Immunoblotting Experiments for MTORC2-AKT Activation (Fig 1).**
(XLSX)

**S3 Data. Data for Relative Accessible Surface Area Analyses (Figs 4 and 5).**
(XLSX)

## Acknowledgments

The authors would like to thank all of the patients and families who participated in this study. The authors would also like to thank Dr. Stephen Cederbaum for his helpful feedback and comments on the manuscript and Mr. Mark Sonneveld for his technical assistance.

## Author Contributions

**Conceptualization:** Aaron D. Besterman, Thorsten Althoff, Mark Nellist, Jeff Abramson, Julian A. Martinez-Agosto.

**Data curation:** Aaron D. Besterman, Thorsten Althoff, Irma Gutierrez-Mejia, Joshua Sadik, Mark Nellist.

**Formal analysis:** Aaron D. Besterman, Thorsten Althoff, Peter Elfferich, Mark Nellist.

**Funding acquisition:** Aaron D. Besterman, Jeff Abramson, Julian A. Martinez-Agosto.

**Investigation:** Aaron D. Besterman, Thorsten Althoff, Peter Elfferich, Jonathan A. Bernstein, Yvette van Ierland, Anja A. Kattentidt-Mouravieva, Mark Nellist, Jeff Abramson, Julian A. Martinez-Agosto.

**Methodology:** Aaron D. Besterman, Thorsten Althoff, Mark Nellist, Jeff Abramson, Julian A. Martinez-Agosto.

**Project administration:** Aaron D. Besterman, Irma Gutierrez-Mejia, Joshua Sadik, Julian A. Martinez-Agosto.

**Resources:** Aaron D. Besterman, Julian A. Martinez-Agosto.

**Supervision:** Mark Nellist, Jeff Abramson, Julian A. Martinez-Agosto.

**Validation:** Aaron D. Besterman, Thorsten Althoff.

**Visualization:** Aaron D. Besterman, Thorsten Althoff.

**Writing – original draft:** Aaron D. Besterman, Thorsten Althoff, Mark Nellist, Julian A. Martinez-Agosto.

**Writing – review & editing:** Aaron D. Besterman, Thorsten Althoff, Jonathan A. Bernstein, Yvette van Ierland, Anja A. Kattentidt-Mouravieva, Mark Nellist, Jeff Abramson, Julian A. Martinez-Agosto.

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
