## [Decision Letter · Decision Letter 0]

12 Mar 2021

Dear Dr Besterman,

Thank you very much for submitting your Research Article entitled 'Functional and Structural Analyses of Novel Smith-Kingsmore Syndrome-Associated MTOR Variants Reveal Potential New Mechanisms and Predictors of Pathogenicity' to PLOS Genetics.

The manuscript was fully evaluated at the editorial level and by independent peer reviewers. The reviewers appreciated the attention to an important problem, but raised some substantial concerns about the current manuscript. Based on the reviews, we will not be able to accept this version of the manuscript, but we would be willing to review a much-revised version. We cannot, of course, promise publication at that time.

If you decide to revise the manuscript for further consideration at PLOS Genetics, please aim to resubmit within the next 60 days, unless it will take extra time to address the concerns of the reviewers, in which case we would appreciate an expected resubmission date by email to plosgenetics@plos.org.

Thorough review has identified a number of issues with the manuscript. In particular reviewers 1 and 4 had major concerns that must be addressed well in a revised manuscript. In addition, all data used to generate the figures must be supplied in a supplemental table.

[LINK]

We are sorry that we cannot be more positive about your manuscript at this stage. Please do not hesitate to contact us if you have any concerns or questions.

Yours sincerely,

David J. Kwiatkowski

Section Editor: Cancer Genetics

PLOS Genetics

Gregory Copenhaver

Editor-in-Chief

PLOS Genetics

Reviewer's Responses to Questions

**Comments to the Authors:**

Reviewer #1: Authors report 7 patients with SKS with 4 novel MTOR variants, describe new phenotypes, confirm MTOR activation of the variants by detecting MTORC1 activation, and describe MTORC2 activation as a new potential mechanism in SKS. Authors found that pathogenic MTOR variants cluster in hotspots in the core of the protein and suggest that 3D properties of MTOR may help in the pathogenicity classification of the variants.

The description of new patients and variants and the functional confirmation of the activity of these variants are of great importance to increase the knowledge in this rare disease with very few cases described to date. Also, the three-dimensional approach as an extra element for assessing a variant's pathogenicity is quite exciting and promising.

However, there are a couple of things in the results that catch my attention. Below I write some questions and comments that arose when reviewing the work.

Minor:

ABSTRACT: Please check the verb tenses in the abstract.

Lines 699-700: Please enter the text for 1G and 1H in alphabetical order, as it appears in the image, to avoid confusion.

Lines 113-115: "Germline mosaic and constitutive mosaic (e.g., present in all cell types but at different frequencies) MTOR variants cause Smith-Kingsmore syndrome (SKS)." The term "germline mosaic" should be changed to "germline". If the authors want, it can also be mentioned that germline variants in SKS often originate from a "germline mosaicism" in one parent. On the other hand, the term "constitutive mosaic" is used in reference 3 to refer to a mosaic mutation present in all cell types but not in 100% of the cells. However, in a recent review and proposal to classify somatic mosaicism (in which many of the authors of reference 3 participate. See PMID: 32661356), it has been preferred to discontinue using the term "constitutional mosaicism" as it can be confusing because the term "constitutive" by itself is used to denote the opposite (non-mosaic variants). I suggest using "somatic mosaic" so that the whole sentence reads: "Germline and somatic mosaic MTOR pathogenic variants cause Smith-Kingsmore syndrome (SKS)".

Major:

In the Results section, the authors say that (lines 184-185) "Unexpectedly, the T389/S6K ratio in the absence of MTOR had an approximately 2-fold increase relative to wild-type MTOR". Later in the Discussion section, the authors say that (lines 298-300) "Unexpectedly, we found that exogenous expression of wild-type MTOR resulted in decreased MTORC1 activity". So, in the results section, the authors seem to assume that the construct without MTOR causes increased activity of MTORC1, while in the discussion section, the authors seem to assume the opposite, that is, it is actually the construct with the wild-type MTOR the one causing a decrease in MTORC1 activity. It is not very clear which of the two occurs, nor do the authors give a plausible explanation that justifies it. They do suggest, without explaining it, that (lines 300-301) "the exogenously expressed MTOR may interfere with the formation of active MTORC1 complexes". If this is so, could it be affecting the constructs with the mutations as well? Would this invalidate the results about MTORC1 activation? The same construct seems not to affect the MTORC2 pathway since the S473/AKT activity is similar with the exogenous wild type MTOR or with the plasmid without MTOR. Please explain.

Lines 195-200: "Except for the p.C1483F variant, we did not observe large differences in the S473/AKT ratio. In the presence of the p.C1390Y and Q2524K variants the S473/AKT ratios were increased approximately 20%...In contrast, the S473/AKT ratios for the p.K1395R and p.V2406M variants were not significantly different from wild-type MTOR". If I have not misinterpreted, these do not appear to correspond to what is shown in figure 1, where (although with a larger variation bar) the p.V2406M variant appears to have a slightly higher ratio than the p.C1390Y variant. Please check to be able to assess these results.

Reviewer #2: This is nice manuscript focusing on the functional aspects of Smith-Kingsmore syndrome and the behavior of mTOR variant proteins.

The paper increases the knowledge of the biology of this important pathway and adds new useful information for the scientific community.

I have some suggestions to improve the MS:

1- Regarding the biology of the variant mTOR protein, more information of the potential treatments and experimental compounds should be incorporated in the text.

2- It would be important for clinicians to include clinical pictures of the patients’ phenotype.

3- The description of the CNS findings, mainly focused in the MRI/CT studies should be discussed in the text.

Minor comments:

The term “1” in number must be replaced by “one” (page 7, l.159 and page 15, l.289).

Reviewer #3: The authors identified 4 novel MTOR mutations from 7 patients sufficing SKS, and tested in vitro the MTOR variant expression levels and substrate phosphorylation ratios of MTORC1 and MTORC2. These parts are interesting and really impressive. However, the correlation between accessible surface area of mutated resides and the quantitative phosphorylation function is questionable. Mutations make effects by affecting the expression levels of the variants or by finely adjusting the relative orientation of N-lobe and C-lobe of kinase domain. Especially, the mutations within the helical interface of FAT domain may stretch or shrink the FAT helical bundles and therefore affect the kinase domain architecture. As a matter of fact, accessible surface area of 3D protein structure is a community effect of all amino acid side chain. Frankly speaking, it is a statistical quality meaningful only for the whole protein or entire interface. Anyway, although the authors have differentiated the RSA with the multi-sample nonparametric test, the linkage seems a little bit weak. A few comments:

1. Page 37, legend for Fig. 1. (H) & (G) should be exchanged.

2. Page 37, legend for Fig. 2. 6BCX & 6BCU are cryo-EM structures, not crystal structures.

3. In the 3.0 A resolution cryo-EM map of 6BCX, the side chain assignment is only confident starting from L599 of N-HEAT domain. Therefore, any discussion about mutations before residue 599 of N-HEAT in Fig 4 is unbelievable.

4. Furthermore, the authors should not trust the coordinate model too much directly downloaded from Protein Data Bank, and should re-examine every important rotamer discussed in the cryo-EM map. For example, side chain of C1390 in Fig. 3A actually is in the same orientation as the mutated Y, if carefully inspected in the cryo-EM map.

Reviewer #4: This manuscript described novel clinical features of patients with SKS and found four pathogenic MTOR variants relevant with MTOR Complex 1/2 activation. The relative accessible surface area was significantly lower for SKS-associated variants

compared to benign variants. With some structural analysis, the author state that these MTOR variants may help in pathogenicity classification. Overall, this manuscript gives some findings for SKS disease, however there are some questions:

1. Figure 1 quality is too low and the enzyme assay is not so good to get a clear result. The S6K and AKT should be the same level if you transfect the same amount for each MTOR variant. It is weird that p-S6K level is increased without transfecting mTOR? How many times was this experiment repeated? At least, the figure 1A/E aren’t so convinced of the result.

2. Figure 2 A/B is almost the same, the quality is too low to see the detailed amino acids. Figure 2 and 3 are analyzed based on the previous structure.

The quality of this manuscript, especially the figures, are not good. There was no obvious novelty of this manuscript. Thus, it is not recommended to be published on the PLOS Genetics journal.

**Have all data underlying the figures and results presented in the manuscript been provided?**

Reviewer #1: **No: **The data associated with the Figures has not been provided

Reviewer #2: Yes

Reviewer #3: Yes

Reviewer #4: Yes

PLOS authors have the option to publish the peer review history of their article (what does this mean?). If published, this will include your full peer review and any attached files.

Reviewer #1: **Yes: **Victor Martinez-Glez

Reviewer #2: No

Reviewer #3: No

Reviewer #4: No

---

## [Decision Letter · Decision Letter 1]

8 Jun 2021

Dear Dr Besterman,

We are pleased to inform you that your manuscript entitled "Functional and Structural Analyses of Novel Smith-Kingsmore Syndrome-Associated MTOR Variants Reveal Potential New Mechanisms and Predictors of Pathogenicity" has been editorially accepted for publication in PLOS Genetics. Congratulations!

Yours sincerely,

David J. Kwiatkowski

Section Editor: Cancer Genetics

PLOS Genetics

Gregory Copenhaver

Editor-in-Chief

PLOS Genetics

Comments from the reviewers (if applicable):

Reviewer's Responses to Questions

**Comments to the Authors:**

Reviewer #1: The authors have responded satisfactorily to the comments and have made the relevant changes, so I consider that the paper is now suitable for publication.

Reviewer #3: The authors have addressed all the comments. I agree it's ready for acceptance.

Reviewer #4: The revised version of the manuscript appropriately addresses all comments and concerns and is suitable for publication in PLOS GENETICS.

**Have all data underlying the figures and results presented in the manuscript been provided?**

Reviewer #1: Yes

Reviewer #3: Yes

Reviewer #4: Yes

PLOS authors have the option to publish the peer review history of their article (what does this mean?). If published, this will include your full peer review and any attached files.

Reviewer #1: **Yes: **Victor Martinez-Glez

Reviewer #3: No

Reviewer #4: No

**Data Deposition**

http://datadryad.org/submit?journalID=pgenetics&manu=PGENETICS-D-21-00280R1

**Press Queries**

---

## [Editor Report · Acceptance letter]

25 Jun 2021

PGENETICS-D-21-00280R1 

Functional and Structural Analyses of Novel Smith-Kingsmore Syndrome-Associated MTOR Variants Reveal Potential New Mechanisms and Predictors of Pathogenicity 

Dear Dr Besterman, 

We are pleased to inform you that your manuscript entitled "Functional and Structural Analyses of Novel Smith-Kingsmore Syndrome-Associated MTOR Variants Reveal Potential New Mechanisms and Predictors of Pathogenicity" has been formally accepted for publication in PLOS Genetics! Your manuscript is now with our production department and you will be notified of the publication date in due course.

With kind regards,

Olena Szabo

PLOS Genetics

On behalf of:
